# Detection of six novel de novo mutations in individuals with low resilience to psychological stress

Esfandiar Azadmarzabadi[1], Arvin Haghighatfard[2,3,4]*

1 Behavioral Sciences Research Center, Life Style Institute, Baqiyatallah University of Medical Sciences, Tehran, Iran, 2 Neuroimaging Genetic Laboratory, Arvin Gene Company, Tehran, Iran, 3 Department of Biology, Tehran North Branch, Islamic Azad University, Tehran, Iran, 4 Department of Genetics, Tehran Medical Sciences Branch, Islamic Azad University, Tehran, Iran

* Arvinland@yahoo.com

**Data Availability Statement:** Stress problem may consider as a big disadvantage for several career. Due to the working and insurance situation of participants who admitted from institutional

## Abstract

Genetic bases of psychological stress resilience have been studied previously, but mechanisms and genetic variants which are involved in stress resilience are still unclear. The present study aimed to evaluate the associations between variants in dopaminergic pathway genes with stress resilience. Subjects of the present study were divided into four groups. Group A included persons with normal reactions to major life events stressors; group B included persons with an acute stress reaction to major life events stressor; group C included persons with normal reactions to Crises/catastrophes stressors, and group D included persons with an acute stress reaction to Crises/catastrophes stressors. DNA was extracted from the subject's blood, and the entire length of 14 genes DRD1, DRD2, DRD3, DRD4, DRD5, COMT, DBH, TH, MAOA, DDC, DAT, 5-HTT, BDNF, and GDNF were sequenced by automated sequencers ABI 3700. Results showed 24 point mutations in 12 genes, including 16 SNPs and six novel mutations, which were significantly correlated to low-stress resilience. Most of the SNPs were known as risk alleles in psychiatric disorders. Several associations were found between genetic variants and psychological characteristics. Findings suggest dopaminergic as an important pathway in stress and stress resilience also indicated shared genetic bases between low-stress resilience and several psychiatric disorders.

## Introduction

Psychological stress occurs when a person perceives that environmental demands exceed his or her adaptive capacity. In these conditions, a person's ability to do the tasks appropriately with minimal anxiety level define as stress resilience [1]. Stressful experiences may lead to major psychiatric problems such as depression, post-traumatic stress disorder (PTSD), and suicidality in susceptible individuals. However, the psychological responses of different persons to the same stressful life events are extremely variable, which could be related to life background and genetic variations [2]. Previous studies determined that environmental, genetic,

**Funding:** Arvin Gene company funded the project and helped us for laboratory process. The funder provided support in the form of salaries for authors, contributor specialists, and materials. Also the neuroimaging genetic laboratory of Arvin gene used for conducting the main part of the study. Arvin Gene company did not have any additional role in the study design, data collection and analysis, decision to publish, or preparation of the manuscript. The specific roles of these authors are articulated in the 'author contributions' section."

**Competing interests:** Arvin Gene company policies does not alter our adherence to PLOS ONE policies on sharing data and materials. Also authors and Arvin Gene company as funder of project declare that they have no conflict of interests.

epigenetic, and neural activities impact resilience, which may mediate by adaptive changes in several neural circuits involving several neurotransmitters and molecular pathways. However, genes, pathways, and biological mechanisms of stress and stress resilience still are not entirely clarified [3]. Detection of risk alleles that are associated with stress resilience may help to predict the vulnerability of persons before they experience stressful conditions and prevention of major psychiatric disorders such as post-traumatic stress disorder (PTSD) and depression, caused by stressful life events.

Previous studies indicate polymorphisms within two key genes, CRHR1 and FKBP5, could be related to stress resilience by the impact on the regulation of the hypothalamic-pituitary-adrenal (HPA) axis function [4]. Several studies suggest that sensitivity to stress-induced anhedonia is associated with the impairment of hippocampal neurogenesis [5, 6]. Dopamine affects several brain processes that control both motor and emotional behavior and plays a role in the brain's reward mechanism. Serotonin is critical in temperature regulation, sensory perception, locomotion, sleep. Dopamine and serotonin systems in the hippocampal, prefrontal cortex, and interconnected neural circuits could be important mechanisms underlying the low-stress resilience and its co-morbid disorders [7].

Caspi et al. reported individuals with one or two copies of the short allele of a functional polymorphism in serotonin transporter(5-HTT) promoter exhibited more depressive symptoms, diagnosable depression, and suicidality symptoms after stressful life events, in comparison with individuals that carry two long alleles [2]. Also, a well-known functional polymorphism called Val66Met in the brain-derived neurotrophic factor (BDNF) gene was found associated with stress vulnerability [8]. Studies that focused on gene-environment interactions in stress resilience reported associations between susceptibility to life stressors and risk alleles, especially 5-HTTLPR and BDNF on depression [9].

The present study aimed to evaluate the role of genetic variations in genes involved in dopamine and serotonin pathways in subjects with low stress resilience. Selected genes were involved in the synthesis, transportation, and degradation of dopamine and serotonin, and most of them implicated as candidate genes in psychiatric disorders such as depression. Fourteen genes which assessed using nucleotide sequencing, include five receptors of dopamine: DRD1(5q35), DRD2(11q23), DRD3(3q13), DRD4(11p15), and DRD5(4p16); five genes which involved in the synthesis and degradation of dopamine: COMT(21q11), DBH(9q34), TH (11p15), MAOA(Xp11) and DDC(7p12); two genes involved in the transportation of dopamine and serotonin: DAT(5p15) and 5-HTT(17q11) and two neurotrophic factors which are targets of dopamine and serotonin: BDNF(11p13) and GDNF(5p13.1-p12). Also, psychological parameters such as personality factors, intelligence, stress, anxiety, depression, and psychological resilience, studied in subjects.

## Material and methods

### Subject selection

The study included Iranian individuals aged 19 to 48 years old who were divided into four groups. Group A included 390 persons with normal reaction to major life events stressors such as high school and university exams, job interviews, sports competitions (218 male, 172 female); group B included 124 persons with an acute and low resilient stress reaction to the same major life events stressor (71 male, 53 female); group C included 240 persons with normal reaction to Crises/catastrophes stressors such as the death of first relatives, divorce or emotional relationships, breakup, major financial problems (165 male,75 female); group D included 117 persons with an acute and low resilient stress reaction to same Crises/catastrophes stressors(86 male,31 female). Subjects were divided into the groups by decision of two

independent senior psychiatrists based on unstructured interviews and results of Depression, Anxiety, Stress (DASS-21), and Connor-Davidson Resilience Scales [10]. Subjects in all groups were matched for sex, age, race, socioeconomic situation, familial situation, and education. Subjects had no history of any psychological or severe somatic problems. Subjects were recruited from psychological outpatient clinics. All subjects have explained the purpose of the study, next a written informed consent has been provided based on the Helsinki declaration of ethics in medical research. The study was approved by the central ethical committee of the Islamic Azad University board including Dr. Faramarz Mohammadi, Dr. Saied Shahsavari, Dr. Laleh Haghparast, Dr. Eshagh Davari, and Dr. Farbod Rezaie (tell: 098-021-44603101 / email: iauakhlaghcomitee@iautnb.ac.ir).

## Analysis of clinical data and psychological assessment

**1) Depression Anxiety Stress Scales (DASS-21).** The DASS is a quantitative measure of distress along the three axes of depression, anxiety, and stress [10]. DASS was constructed to further the process of defining, understanding, and measuring the ubiquitous and clinically significant emotional states, usually described as depression, anxiety, and stress. Each of the three DASS scales contains 14 items, divided into subscales of 2–5 items with similar content. Scores for Depression, Anxiety, and Stress are calculated by summing the scores for the relevant items [10].

**2) Hamilton Anxiety Rating Scale IVR (HAM-A).** The HAM-A was one of the first rating scales developed to measure the severity of anxiety symptoms, and is still widely used today in both clinical and research settings. The scale consists of 14 items, each defined by a series of symptoms, and measures both psychic anxiety (mental agitation and psychological distress) and somatic anxiety (physical complaints related to anxiety) [11].

**3) Hamilton Depression Rating Scale (HDRS-17).** The HDRS is one of the most reliable and widely used clinician-administered depression assessment scales. The original version contains 17 items (HDRS 17) pertaining to symptoms of depression experienced over the past week [12].

**4) Connor-Davidson Resilience Scale (CD-RISC).** Resilience is well known as a measure of stress coping ability. The Connor-Davidson Resilience Scale (CD-RISC) comprises 25 items, each rated on a 5-point scale (0–4), with higher scores reflecting greater resilience. The scale demonstrates that resilience is modifiable and can improve with treatment [13].

**5) Wechsler Adult Intelligence Scale (WAIS-IV).** The Wechsler Adult Intelligence Scale (WAIS) is a test designed to measure intelligence in adults and older adolescents. Verbal working memory and Spatial working memory were measured by subtests of WAIS, Digit span, and dot span [14].

**6) NEO Five-Factor Inventory (NEO-FFI).** The Revised NEO Personality Inventory is a psychological personality inventory, consists of 240 questions intended to measure the Big Five personality traits: Extraversion, Agreeableness, Conscientiousness, Neuroticism, and Openness to Experience. A shortened version, the NEO-FFI, which is used in the present study, contains 60 items (12 items per domain) [15].

## Blood sampling and DNA extraction

Blood (5 ml) was collected from the cubital vein without a tourniquet. Genomic DNA was extracted from peripheral blood samples according to standard protocols using the Genomic DNA Purification Kit (Thermo Fisher Scientific #K0512). The quality and integrity of extracted DNA were evaluated by agarose gel electrophoresis and UV-spectroscopy.

## PCR amplification and DNA sequencing

The entire length of each gene, including coding and non-coding regions, was amplified by PCR, and DNA cycle sequencing on automated sequencers ABI 3700 was conducted as described in previous studies [16]. Parents of subjects, who carry novel mutations on their genome, were examined for the presence of these point mutations by using tetra-primer ARMS-PCR according to the standard protocols using by PCR Master Mix kit (Thermo Fisher Scientific # K0172) and 96-well C1000 Touch thermal cycler (BIO-RAD, California, United States).

## Sequence data and statistical analysis

All of the sequenced data were compared between individuals in groups (A vs. B and C vs. D) by an optimized version of Phred software to ABI 3700, Phred version (0.020425.c). Hardy-Weinberg equilibrium (HWE) was tested using exact significance as implemented in STATA 12.1. Testing of genotypes HWE in all subjects with normal resilience (group A and C) were determined and the threshold for significant deviation from HWE was set as 0.01. Single nucleotide polymorphisms that were fulfilling HWE were included in further analyses. Minor allele frequencies were measured using STATA 12.1. The normality of residuals was checked graphically with STATA 12.1. Linkage disequilibrium (LD) statistics D' and r2 in paired SNPs were calculated using Pairwise LD in PLINK (r2 ≥0.8, D' = 1). For statistical analysis, all descriptive data were expressed as mean ± Standard Deviation. Differences in means between groups were considered significant if p<0.05. Chi-square test used for the detection of group differences in allele frequency and independent t-test. One-way ANOVA was used for the comparison of genetic variants with demographic and psychological data between groups. Multiple-comparison analysis correction was conducted by the Bonferroni correction test.

# Results

## Identification of mutations

Several genetic variations were detected in 14 genes, but most of them were not significant after statistical examinations and Bonferroni correction. The numbers of all variations are provided in Table 1. Genotype proportions were all in HWE for significant SNPs (P> 0.01). After Bonferroni correction, eighteen single nucleotide polymorphisms (SNPs) and six novel point mutations reached significant association with low resilience to stress in group B vs. group A. Sixteen SNPs and six novel point mutations were found significantly related to low resilience to stress in group C vs. group D (Tables 2–4,). Detected novel point mutations were not reported based on NCBI/Gene bank and were present in individuals of both groups with low-stress resilience (B and D). From six novel mutations, two mutations were detected in COMT, one mutation in DRD2, one mutation in GDNF, and two mutations were detected in the 5-HTT gene. In trio strategy, tetra-primer ARMS-PCR for parents of subjects who carried novel mutations showed none of these mutations were present in parents, and all of them were de novo mutations. All detected SNPs were in linkage equilibrium.

## DASS-21 results and correlation with genetic variations

All demographic and clinical tests results are presented in Table 5. Significant correlation between rs25531 in 5-HTT (P = 0.002) with higher depression scale and rs6265 in BDNF (P = 0.002) and higher stress scale detected in all recruited samples. In low resilient groups there were more correlations. There were significant correlations between presence of rs6265 in BDNF (P = 0.002) and rs25531 in 5-HTT (P = 0.003) with higher depression scale in group

**Table 1. Detected genetic variations.**

| Gene | NCBI Reference Sequence accession number | All detected variations | Significant variations in group B vs. A | Significant variations in group D vs. C |
|------|------------------------------------------|-------------------------|------------------------------------------|------------------------------------------|
| DRD1 | NC_000005.10 | 21 | 2 | 1 |
| DRD2 | NC_000011.10 | 24 | 2 | 1 |
| DRD3 | NC_000003.12 | 15 | 1 | 1 |
| DRD4 | NC_000011.10 | 17 | 1 | 1 |
| DRD5 | NC_000004.12 | 7 | 1 | 0 |
| DBH | NC_000009.12 | 14 | 2 | 2 |
| COMT | NC_000022.11 | 18 | 4 | 4 |
| BDNF | NC_000011.10 | 9 | 3 | 3 |
| 5-HTT | NC_000017.11 | 16 | 3 | 3 |
| GDNF | NC_000005.10 | 8 | 2 | 2 |
| DAT | NC_000005.10 | 12 | 2 | 2 |
| TH | NC_000011.10 | 11 | 0 | 0 |
| MAOA | NC_000023.11 | 15 | 1 | 1 |
| DDC | NC_000007.14 | 6 | 0 | 0 |

B. Significant correlation between presence of rs6265 in BDNF (P = 0.003), rs25531 in 5-HTT (P = 0.003), rs1800955 in DRD4 (P = 0.003) and rs1611115 in DBH (P = 0.001) with higher depression scale in group D was detected. Correlation between presence of rs5906957 in MAOA (P = 0.004) with higher anxiety scale in group B was determined. Presence of rs5906957 in MAOA (P = 0.005), rs25531 in 5-HTT (P = 0.007) and rs4680 in COMT (P = 0.001) were associated with higher anxiety scale in group D. There was significant correlation between presence of rs1076560 in DRD2 (P = 0.002), rs4680 in COMT (P = 0.003) and rs6265 in BDNF (P = 0.006) with higher stress scale in group B. Significant correlation was detected between presence of rs4680 in COMT (P = 0.001) and rs6265 in BDNF (P = 0.003) and higher stress scale in group D.

## HAM-A results and correlation with genetic variations

No significant correlation was found in all recruited samples. On the other hand, there was a significant correlation between the presence of rs5906957 in MAOA (P = 0.003) and rs25531 in 5-HTT (P = 0.001) with a higher HAM-A scale in group B. Presence of rs5906957 in MAOA (P = 0.002), rs25531 in 5-HTT (P = 0.001) and rs4680 in COMT (P = 0.002) were associated with higher HAM-A scale in group D.

## HDRS-17 results and correlation with genetic variations

Significant correlation between rs25531 in 5-HTT (P = 0.003) with higher depression scale in HDRS-17 test detected in all recruited samples. There was significant correlation between presence of rs1800955 in DRD4 (P = 0.001) and rs25531 in 5-HTT (P = 0.003) with higher depression scale in group B. Also significant correlation was detected between presence of rs25531 in 5-HTT (P = 0.002), rs1800955 in DRD4 (P = 0.001) and rs1611115 in DBH (P = 0.004) with higher HDRS-17 scale in group D.

## CD-RISC results and correlation with genetic variations

In all recruited samples rs4680 in COMT (P = 0.004) and rs6265 in BDNF (P = 0.005) were significantly detected with low CD-RISC score. There was significant correlation between rs4680 in COMT (P = 0.002) and rs6265 in BDNF (P = 0.003) with decrease in CD-RISC score in

**Table 2. Details and allele frequencies of detected variants and mutations with a significant relation to low stress resilience.**

| No. | Gene | SNP number | Nucleotide substitution | Functional Consequence | Number of persons with minor allele in group A | Number of persons with minor allele in group B | Number of persons with minor allele in group C | Number of persons with minor allele in group D |
|---|---|---|---|---|---|---|---|---|
| 1 | DRD1 | rs548677242 | C/T | Glu ⇒ Lys | 57(15%) | 76(60%) | 32(13%) | 45(39%) |
| 2 | DRD1 | rs779186397 | C/T | Arg ⇒ Lys | 34(9%) | 76(60%) | 38(16%) | 76(66%) |
| 3 | DRD2 | rs1076560 | A/C | Intron variant | 23(6%) | 68(55%) | 43(18%) | 58(49.5%) |
| 4 | DRD2 | Novel mutation | T/C | Promoter | 1(0.25%) | 7(5.6%) | 1(0.41%) | 3(2.5%) |
| 5 | DRD3 | rs6280 | C/T | Ser⇒ Gly | 43(11%) | 57(46%) | 30 (12%) | 22(19%) |
| 6 | DRD4 | rs1800955 | C/T | Promoter | 16(4%) | 59(48%) | 14(6%) | 59(50%) |
| 7 | DRD5 | rs2867383 | A/G | intron variant | 19(5%) | 80(65%) | 17(7%) | 33(28%) |
| 8 | DBH | rs2283123 | C/T | intron variant | 31(8%) | 73(59%) | 5(2%) | 58(50%) |
| 9 | DBH | rs1611115 | C/T | Upstream variant | 70(18%) | 86(69%) | 48(20%) | 71(61%) |
| 10 | COMT | rs165599 | A/G | Intron variant | 42(11%) | 66(53%) | 18(7%) | 73(62%) |
| 11 | COMT | rs4680 | G/A | Val⇒ Met | 12(3%) | 86(70%) | 10(4%) | 46(39%) |
| 12 | COMT | Novel mutation | G/A | Promoter | 1(0.25%) | 6(4.8%) | 1(0.41%) | 4(3.4%) |
| 13 | COMT | Novel mutation | G/T | Promoter | 2(0.5%) | 13(10%) | 1(0.41%) | 8(7%) |
| 14 | MAOA | rs5906957 | A/G | Intron variant | 47(12%) | 57(45%) | 30(12.5%) | 34(29%) |
| 15 | BDNF | rs6265 | A/G | Val⇒ Met | 16(4%) | 68(55%) | 14(6%) | 61(52%) |
| 16 | BDNF | rs146354977 | C/T | Val⇒ Met | 27(7%) | 55(44%) | 22(9%) | 56(48%) |
| 17 | BDNF | rs760902255 | T/C | Asn⇒ Asn | 31(8%) | 58(48%) | 9(4%) | 53(45%) |
| 18 | GDNF | rs752541985 | C/T | Lys ⇒ Arg | 16(4%) | 88(71%) | 17(7%) | 82(70%) |
| 19 | GDNF | Novel mutation | A/T | Lys ⇒Asn | 1(0.25%) | 11(6.4%) | 1(0.41%) | 3(2.5%) |
| 20 | 5-HTT | Novel mutation | C/G | Arg ⇒ Pro | 2(0.5%) | 14(10.8%) | 1(0.41%) | 8(7%) |
| 21 | 5-HTT | Novel mutation | C/G | Ala⇒ Pro | 1(0.25%) | 10(8%) | 1(0.41%) | 5(4.2%) |
| 22 | 5-HTT | rs25531 | A/G | Intron variant | 32(8%) | 59(48%) | 16(7%) | 72(62%) |
| 23 | DAT | rs431905515 | C/T | Leu ⇒ Pro | 16(4%) | 98(79%) | 19(8%) | 83(71%) |
| 24 | DAT | rs431905516 | C/T | Arg ⇒ Trp | 17(4%) | 89(72%) | 26(11%) | 80(69%) |

mut: mutation, SNP: single nucleotide polymorphism, Chr: chromosome.

group B. Also associations between rs4680 in COMT (P = 0.001) and rs6265 in BDNF (P = 0.003) and decrease in CD-RISC score were detected in group D.

## WAIS-IV results and correlation with genetic variations

No significant correlation was observed for the total IQ score in WAIS-IV, in all samples or each group. The only significant correlation was found between rs4680 in COMT (P = 0.001) and a decrease in dot spans score detected in group B.

## NEO-FFI results and correlation with genetic variations

Results showed higher neuroticism and lower extraversion total scores in groups B and D compared with groups A and C, respectively. A significant correlation was observed between a higher neuroticism score and a decrease in CD-RISC score in all 871 recruited subjects (P = 0.003). In addition, a significant correlation was detected between the presence of

**Table 3. Genotype frequencies of genetic variants associated with low resilience genotyping.**

| SNP number | Genotypes | Group A 390 | Group B 124 | Group C 240 | Group D 117 |
|---|---|---|---|---|---|
| rs548677242 | CC | 333 (85%) | 48(38.7%) | 208(86.6%) | 72(61.5%) |
| | TT | 17 (4.3%) | 64(51.6%) | 14(5.8%) | 10(8.5%) |
| | CT | 40 (10.7%) | 12(9.7%) | 18(7.5%) | 35(29.9%) |
| rs779186397 | CC | 356(91.2%) | 48(38.7%) | 202(84%) | 41(35%) |
| | TT | 22(5.6%) | 61(49%) | 29(12%) | 56(47.8%) |
| | CT | 12(3%) | 15(12%) | 9(3.7%) | 20(17%) |
| rs1076560 | CC | 367(94%) | 56(45%) | 197 (82%) | 59(50.4%) |
| | AA | 18(4.6%) | 64(51.6%) | 38 (15.8%) | 47(40%) |
| | AC | 5(1.28%) | 4(3.2%) | 5(2%) | 11(9.4%) |
| Novel mutation of DRD2 | CC | 389(99.7%) | 116(93.5%) | 239(99.59%) | 114(97.5%) |
| | TT | 1(0.25%) | 7(5.6%) | 1(0.41%) | 3(2.5%) |
| | CT | 0(0%) | 0(0%) | 0(0%) | 0(0%) |
| rs6280 | CC | 347(88.9%) | 67(54.3%) | 210(87.5%) | 95(81%) |
| | TT | 40(10.2%) | 52(42%) | 26 (10.8%) | 21(18%) |
| | CT | 3(0.76%) | 5(4%) | 4(1.6%) | 1(0.8%) |
| rs1800955 | TT | 374 (95.8%) | 65(52.4%) | 226(94%) | 58(49.5%) |
| | CC | 12(3.07%) | 54(43.5%) | 9(3.7%) | 47(40%) |
| | CT | 4(1.02%) | 5(4.03%) | 5(2%) | 12(10%) |
| rs2867383 | GG | 371 (95.1%) | 66(53.2%) | 223(93%) | 84(71.7%) |
| | AA | 12(3.07%) | 66(53.2%) | 8(3.3%) | 26(22.2%) |
| | AG | 7(1.79%) | 14(11.2%) | 9(3.7%) | 7(6%) |
| rs2283123 | CC | 359(92.05%) | 51(41%) | 235(98%) | 59(50.4%) |
| | TT | 23(5.8%) | 68(54.8%) | 1(0.41%) | 48(41%) |
| | CT | 8(2.05%) | 5(4%) | 4(1.6%) | 10(8.5%) |
| rs1611115 | CC | 320(82.05%) | 38(30.6%) | 192(80%) | 46(39%) |
| | TT | 54(13.8%) | 75(60.4%) | 29(12%) | 65(55.5%) |
| | CT | 16(4.1%) | 11(8.8%) | 19(7.9%) | 6(5%) |
| rs165599 | AA | 354(90.7%) | 58(46.7%) | 222 (92.5%) | 44(37.6%) |
| | GG | 15(3.08%) | 45(36.2%) | 6(2.5%) | 55(47%) |
| | AG | 21(5.3%) | 21(17%) | 12(5%) | 18(15.3%) |
| rs4680 | GG | 378(96.9%) | 38(30.6%) | 224(93.3%) | 71(60.6%) |
| | AA | 2(0.5%) | 78(63%) | 10(4.1%) | 42(36%) |
| | AG | 10(2.5%) | 8(6.4%) | 6(2.5%) | 4(3.4%) |
| First novel Mutation of COMT | AA | 1(0.25%) | 6(4.8%) | 1(0.41%) | 4(3.4%) |
| | GG | 389(99.7%) | 98(79.3%) | 239(99.5%) | 113(96.6%) |
| | AG | 0(0%) | 0(0%) | 0(0%) | 0(0%) |
| Second novel mutation of COMT | GG | 388(99.4%) | 101(81.4%) | 239(99.5%) | 109(93%) |
| | TT | 2(0.5%) | 13(10%) | 1(0.41%) | 8(6.8%) |
| | GT | 0(0%) | 0(0%) | 0(0%) | 0(0%) |
| rs5906957 | AA | 343(87.9%) | 67 (54%) | 210(87.5%) | 83(71%) |
| | GG | 34(8.7%) | 47(38%) | 26(11%) | 29(24.8%) |
| | AG | 13(3.3%) | 10(8%) | 4(1.6%) | 5(4.2%) |
| rs6265 | GG | 374(95.8%) | 56(45%) | 226(94%) | 56(47.8%) |
| | AA | 13(3.3%) | 53(42.7%) | 9(3.7%) | 49(41.8%) |
| | AG | 3(0.76%) | 15(12%) | 5(2%) | 12(10%) |

(*Continued*)

**Table 3.** (Continued)

| SNP number | Genotypes | Group A 390 | Group B 124 | Group C 240 | Group D 117 |
|---|---|---|---|---|---|
| rs146354977 | CC | 363(93.07%) | 69(55.6%) | 218(91%) | 61(52%) |
| | TT | 12(3.07%) | 39(31.4%) | 14(5.8%) | 48(41%) |
| | CT | 15(3.8%) | 16(13%) | 8(3.3%) | 8(6.8%) |
| rs760902255 | CC | 359(92%) | 66(53.2%) | 231(96.2%) | 64(54.7%) |
| | TT | 25(6.4%) | 55(44.3%) | 7(3%) | 47(40%) |
| | CT | 6(1.5%) | 3(2.4%) | 2(0.8%) | 6(5.1%) |
| rs752541985 | TT | 374(95.8%) | 36(29%) | 223(93%) | 35(30%) |
| | CC | 13(3.3%) | 80(64.5%) | 11(4.5%) | 65(55.5%) |
| | CT | 3(0.7%) | 8 (6.4%) | 6(2.5%) | 17(14.5%) |
| Novel mutation of GDNF | TT | 389(99.75%) | 113(91.2%) | 239(99.59%) | 114(97.5%) |
| | AA | 1(0.25%) | 10(8%) | 1(0.41%) | 3(2.5%) |
| | AT | 0(0%) | 1(0.8%) | 0(0%) | 0(0%) |
| First novel mutation of 5-HTT | CC | 388(99.5%) | 110(88.7%) | 239(99.59%) | 109(93%) |
| | GG | 2(0.5%) | 14(11.3%) | 1(0.41%) | 8(7%) |
| | CG | 0(0%) | 0(0%) | 0(0%) | 0(0%) |
| Second novel mutation of 5-HTT | CC | 389(99.75%) | 114(92%) | 239(99.59%) | 112(95.8%) |
| | GG | 1(0.25%) | 10(8%) | 1(0.41%) | 5(4.2%) |
| | CG | 0(0%) | 0(0%) | 0(0%) | 0(0%) |
| rs25531 | AA | 358(91.7%) | 65(52.4%) | 224(93.3%) | 45(38.4%) |
| | GG | 27(7%) | 44(35.4%) | 5(2%) | 63(53.8%) |
| | AG | 5(1.2%) | 15(12%) | 11(4.5%) | 9(7.6%) |
| rs431905515 | TT | 374(95.8%) | 26(21%) | 221(92%) | 34(29%) |
| | CC | 4(1.02%) | 83(67%) | 10(4.1%) | 63(53.8%) |
| | CT | 12(3.07%) | 15(12%) | 9(3.7%) | 20(17%) |
| rs431905516 | CC | 373(95.6%) | 35(28%) | 214(89%) | 37(31.6%) |
| | TT | 6(1.5%) | 67(54%) | 18(7.5%) | 63(53.8%) |
| | CT | 11(2.8%) | 22(17.7%) | 8(3.3%) | 17(14.5%) |

rs5906957 in MAOA and higher neuroticism scores in group B (P = 0.002) and group D (P = 0.002). Also, a significant correlation was found between rs5906957 in MAOA and higher neuroticism scores in all samples together (P = 0.005). Statistical analysis results for demographic and clinical characteristics between groups are presented in Table 6.

## Discussion

Detected SNPs and novel mutations were located on 12 genes that are involved in the dopaminergic pathway. Two SNPs were detected in the DRD1 gene. Genetic variations of DRD1 are associated with schizophrenia, aggression, and psychosis symptoms of Alzheimer patients, but detected SNPs in the present study were not detected in any disorder or behavior before [17, 18]. Two SNPs were detected in the DRD2 gene. DRD2 is an important gene in the dopamine pathway, and genetic variations of DRD2 are involved in schizophrenia and susceptibility to post-traumatic stress disorder. Association of rs1076560 in DRD2, which were associated with low-stress resilience, previously was reported to influence memory, alcoholism and modulate the risk of opiate addiction and the dosage requirements of methadone substitution [17, 19]. The location of the novel mutation that was detected in DRD2 is in the promoter region, and as this mutation is de novo, it may change the expression regulation of the gene and cause low resilience to stress. DRD3, DRD4, and DRD5 showed three significantly associated SNPs to

**Table 4. Statistical analysis results of genetic variants associated with low resilience.**

| No. | Gene | SNP number | B vs. A | D vs. C | HWE | Pc Value |
|-----|------|-----------|---------|---------|-----|----------|
| 1 | DRD1 | rs548677242 | P = 0.004 | P = 0.12 | 0.26 | 0.0083 |
| 2 | DRD1 | rs779186397 | P = 0.002 | P = 0.003 | 0.14 | 0.0083 |
| 3 | DRD2 | rs1076560 | P = 0.003 | P = 0.087 | 0.37 | 0.0083 |
| 4 | DRD2 | Novel mutation | P = 0.004 | P = 0.002 | - | 0.0083 |
| 5 | DRD3 | rs6280 | P = 0.002 | P = 0.003 | 0.18 | 0.016 |
| 6 | DRD4 | rs1800955 | P = 0.003 | P = 0.001 | 0.29 | 0.016 |
| 7 | DRD5 | rs2867383 | P = 0.002 | P = 0.16 | 0.43 | 0.016 |
| 8 | DBH | rs2283123 | P = 0.001 | P = 0.003 | 0.22 | 0.0083 |
| 9 | DBH | rs1611115 | P = 0.001 | P = 0.002 | 0.45 | 0.0083 |
| 10 | COMT | rs165599 | P = 0.004 | P = 0.003 | 0.36 | 0.0041 |
| 11 | COMT | rs4680 | P = 0.002 | P = 0.004 | 0.2 | 0.0041 |
| 12 | COMT | Novel mutation | P = 0.003 | P = 0.001 | - | 0.0041 |
| 13 | COMT | Novel mutation | P = 0.001 | P = 0.003 | - | 0.0041 |
| 14 | MAOA | rs5906957 | P = 0.004 | P = 0.002 | 0.19 | 0.016 |
| 15 | BDNF | rs6265 | P = 0.002 | P = 0.003 | 0.26 | 0.0055 |
| 16 | BDNF | rs146354977 | P = 0.001 | P = 0.001 | 0.65 | 0.0055 |
| 17 | BDNF | rs760902255 | P = 0.003 | P = 0.001 | 0.48 | 0.0055 |
| 18 | GDNF | rs752541985 | P = 0.002 | P = 0.001 | 0.33 | 0.0081 |
| 19 | GDNF | Novel mutation | P = 0.003 | P = 0.003 | - | 0.0081 |
| 20 | 5-HTT | Novel mutation | P = 0.002 | P = 0.002 | - | 0.0055 |
| 21 | 5-HTT | Novel mutation | P = 0.002 | P = 0.002 | - | 0.0055 |
| 22 | 5-HTT | rs25531 | P = 0.001 | P = 0.003 | 0.25 | 0.0055 |
| 23 | DAT | rs431905515 | P = 0.001 | P = 0.002 | 0.72 | 0.008 |
| 24 | DAT | rs431905516 | P = 0.003 | P = 0.001 | 0.3 | 0.008 |

SNP: single nucleotide polymorphism, HWE: Hardy-Weinberg Equilibrium, *Pc value: P-value after Bonferroni correction.

**Table 5. Demographic and clinical characteristics in groups.**

| variables | Group A | Group B | Group C | Group D |
|-----------|---------|---------|---------|---------|
| Gender | 218 male | 71 male | 165 male | 86 male |
| | 172 female | 53 female | 75 female | 31 female |
| Age | 34±12 | 32±14 | 33±8 | 33±11 |
| Depression(DASS-21) | 5±1.6 | 22±2.3 | 7±1.1 | 28±3.3 |
| Anxiety (DASS-21) | 4.1±1.8 | 23±2.5 | 6.2±1.4 | 32±3.3 |
| Stress (DASS-21) | 9±3.3 | 32±0.8 | 14±1.1 | 37±1.6 |
| HAM-A | 17±2 | 29±4 | 17±5 | 32±4 |
| HDRS-17 | 6±1 | 34±5 | 12±5 | 45±1 |
| CD-RISC | 74±16 | 61±4 | 71±2 | 68±5 |
| IQ total | 107±32 | 97±5 | 96±21 | 93±11 |
| Neuroticism | 38±7 | 61±2 | 44±5 | 68±4 |
| Extraversion | 59±3 | 50±4 | 55±3 | 47±6 |
| Openness | 54±8 | 45±6 | 51±7 | 52±4 |
| Agreeableness | 56±4 | 55±4 | 46±2 | 51±6 |
| Conscientiousness | 56±5 | 48±4 | 49±3 | 52±7 |

DASS-21: Depression Anxiety Stress Scales, HAM-A: Hamilton Anxiety Rating Scale, HDRS-17: Hamilton Depression Rating Scale, CD-RISC: Connor-Davidson Resilience Scale.

**Table 6. Statistical analysis results for demographic and clinical characteristics between groups.**

| variables | Group A vs. B | Group C vs. D |
|---|---|---|
| Gender | p value: 0.93 | p value: 0.88 |
| Age | p value: 0.98 | p value: 1.22 |
| Depression(DASS-21) | p value: 0.003* | p value: 0.002* |
| Anxiety (DASS-21) | p value: 0.003* | p value: 0.003* |
| Stress (DASS-21) | p value: 0.004* | p value: 0.002* |
| HAM-A | p value: 0.003* | p value: 0.003* |
| HDRS-17 | p value: 0.002* | p value: 0.004* |
| CD-RISC | p value: 0.003* | p value: 0.003* |
| IQ total | p value: 0.15 | p value: 0.28 |
| Neuroticism | p value: 0.003* | p value: 0.004* |
| Extraversion | p value: 0.003* | p value: 0.004* |
| Openness | p value: 0.11 | p value: 0.17 |
| Agreeableness | p value: 0.14 | p value: 0.22 |
| Conscientiousness | p value: 0.21 | p value: 0.19 |

DASS-21: Depression Anxiety Stress Scales, HAM-A: Hamilton Anxiety Rating Scale, HDRS-17: Hamilton Depression Rating Scale, CD-RISC: Connor-Davidson Resilience Scale.

*: Significance.

low stress resilience. SNP of DRD4 in the promoter may influence in expression regulation of the gene. DBH is an important part of the dopaminergic pathway. Several genetic variations and SNPs in DBH are associated with psychiatric disorders such as ADHD. Two SNPs detected in this gene are associated with ADHD, but this is the first report of the association of these SNPs with low resilience to stress [20]. Four SNPs and novel mutations were detected in COMT. COMT is one of the most important genes associated with behavioral properties and psychotic disorders. Genetic variant rs4680 (Val158Met) in COMT is associated with schizophrenia and personality disorders [21]. Novel mutation's location in COMT is in the promoter region that could change in expression regulation and effects on the degradation of dopamine. One SNP was detected in MAOA. Association of this SNP (rs5906957) with anger and ADHD had been reported [22]. BDNF is an important neurotrophic factor with the leading role in the regulation of the different parts of behavior. Three SNPs in this gene were detected in association with low stress resilience, including rs6265. Previously correlation of rs6265 and rs4680 in COMT with childhood trauma was reported [23]. In the present study, the correlation of rs6265 and rs4680 with low stress resilience was detected. Three mutations found in BDNF are near, and this region could be a hot spot region for low stress resilience. GDNF is another important neurotrophic factor with a great impact on behavior. A previous study reported rs752541985 may be associated with Hirschsprung disease [24]. Associations of rs752541985 and one novel mutation in GDNF to low stress resilience were detected. 5-HTT is the most well-known gene in the genetic of stress. Caspi et al in 2003 detected variations in this gene which were involved in stress response [6]. Three variants in 5-HTT, including two novel mutations, were detected in the present study. The functional consequences of both novel mutations in 5-HTT were the substitution of Proline that can break the polypeptide chain. Polypeptide chain break, in turn, may lead to dysfunction of serotonin transportation. Dopamine and 5-hydroxytryptamine are both formed as reciprocal intrarenal hormones by the aromatic L-amino-acid decarboxylase enzyme [25], and the main role of 5-HT(1A) receptors in reuptake inhibition and enhancement of 5-HT and DA transmission in the prefrontal cortex

were reported [26]. It seems that the 5-HTT variants could lead to severe deregulation of dopamine signaling in stress-resilient subjects. DAT has a critical role in the transportation of dopamine in the brain. Two SNPs in this gene were detected which were previously reported as pathogen mutations in Infantile Parkinsonism-dystonia. These findings may relate to shared genetic bases of low resilience and Parkinson [27].

Previously in 2018, we studied the expression level of the same 14 genes (dopaminergic signaling pathway genes) considered in the present study, in blood samples of the subjects with normal and low-stress resilience. Results of that study indicate overexpression of DRD1, DRD2, DRD3, DRD4, DBH, DAT, and BDNF as well as the down expression of 5-HTT, MAOA, and COMT [26]. Several possible associations may exist between genetic variants which were found in the present study and expression alterations in these genes. It seems that all together, detected SNPs in the present study may lead to dopamine up-regulation that is related to high anxiety and low resilience [28].

Psychological assessments and their correlations with genetic variations showed that these genetic variants are involved in several behaviors and psychological properties such as personality, memory, anxiety, and depression as well as stress resilience. The results of the present study showed the role of dopaminergic genes on one of the most basic behaviors of humans, stress resilience. Correlation studies of genetic variations with accredited psychological tests make the results more valuable. On the other hand, there were some limitations in our study. We were faced with limitations such as low sample size, the controversy of group definitions, and the absence of neuroimaging data.

After all, it seems that genetic bases of stress resilience deficits as a risk factor for several psychiatric disorders have not been studied enough. Also, further genome-wide association studies and whole genomic sequencing assessments may suggest more shared genetic bases of stress resilience and psychiatric disorders and may help to the prognosis of susceptible persons to low-stress resilience and may prevent them from affecting major psychiatric disorders like PTSD, depression, and schizophrenia.

## Acknowledgments

We should thank all psychiatrists and psychologists who participated in the clinical part of the study. Also, we should thank the subjects and their families for their patience during psychological assessments. We would like to thanks Fatemeh Mohammadpour and Zeinab Tabrizi, Arvin-Gene-Company staff for their valuable participation in the laboratory process.

## Author Contributions

**Conceptualization:** Esfandiar Azadmarzabadi, Arvin Haghighatfard.

**Data curation:** Esfandiar Azadmarzabadi.

**Investigation:** Esfandiar Azadmarzabadi, Arvin Haghighatfard.

**Methodology:** Arvin Haghighatfard.

**Project administration:** Arvin Haghighatfard.

**Resources:** Esfandiar Azadmarzabadi.

**Supervision:** Arvin Haghighatfard.

**Writing – original draft:** Arvin Haghighatfard.

**Writing – review & editing:** Arvin Haghighatfard.

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
