## [Decision Letter · Decision Letter 0]

18 Mar 2021

PONE-D-20-40138

Detection of six novel de novo mutations in individuals with low resilience to psychological stress

PLOS ONE

Dear Dr. Haghighatfard,

Thank you for submitting your manuscript to PLOS ONE. After careful consideration, we feel that it has merit but does not fully meet PLOS ONE’s publication criteria as it currently stands. Therefore, we invite you to submit a revised version of the manuscript that addresses the points raised during the review process.

We look forward to receiving your revised manuscript.

Kind regards,

Weihua YUE, M.D.

Academic Editor

PLOS ONE

Additional Editor Comments (if provided):

Dear Dr. Arvin Haghighatfard,

Thank you for submitting the manuscript entitled, "Detection of six novel de novo mutations in individuals with low resilience to psychological stress" (Manuscript ID: PONE-D-20-40138) to PLOS ONE. The manuscript has now been reviewed, and we think that the manuscript should not be published in the current version, unless you revised it appropriately.

The comments of the reviewers are enclosed for your consideration. We hope the information provided by the reviewers will be helpful.

Yours sincerely,

Weihua Yue,

Academic Editor of PLOS ONE.

Journal Requirements:

2) Thank you for stating the following in the Financial Disclosure section:

[Arvin Gene company funded the project and helped us for laboratory process].   

We note that one or more of the authors are employed by a commercial company: Arvin Gene Company

i.  Please provide an amended Funding Statement declaring this commercial affiliation, as well as a statement regarding the Role of Funders in your study. If the funding organization did not play a role in the study design, data collection and analysis, decision to publish, or preparation of the manuscript and only provided financial support in the form of authors' salaries and/or research materials, please review your statements relating to the author contributions, and ensure you have specifically and accurately indicated the role(s) that these authors had in your study. You can update author roles in the Author Contributions section of the online submission form.

ii. Please also provide an updated Competing Interests Statement declaring this commercial affiliation along with any other relevant declarations relating to employment, consultancy, patents, products in development, or marketed products, etc.  

3)  We note that you have indicated that data from this study are available upon request. PLOS only allows data to be available upon request if there are legal or ethical restrictions on sharing data publicly. For information on unacceptable data access restrictions, please see http://journals.plos.org/plosone/s/data-availability#loc-unacceptable-data-access-restrictions.

4) PLOS requires an ORCID iD for the corresponding author in Editorial Manager on papers submitted after December 6th, 2016. Please ensure that you have an ORCID iD and that it is validated in Editorial Manager. To do this, go to ‘Update my Information’ (in the upper left-hand corner of the main menu), and click on the Fetch/Validate link next to the ORCID field. This will take you to the ORCID site and allow you to create a new iD or authenticate a pre-existing iD in Editorial Manager. Please see the following video for instructions on linking an ORCID iD to your Editorial Manager account: https://www.youtube.com/watch?v=_xcclfuvtxQ

Reviewers' comments:

Reviewer's Responses to Questions

**Comments to the Author**

1. Is the manuscript technically sound, and do the data support the conclusions?

Reviewer #1: Partly

Reviewer #2: No

Reviewer #3: Yes

2. Has the statistical analysis been performed appropriately and rigorously? 

Reviewer #1: Yes

Reviewer #2: Yes

Reviewer #3: Yes

3. Have the authors made all data underlying the findings in their manuscript fully available?

Reviewer #1: Yes

Reviewer #2: Yes

Reviewer #3: No

4. Is the manuscript presented in an intelligible fashion and written in standard English?

Reviewer #1: Yes

Reviewer #2: Yes

Reviewer #3: Yes

5. Review Comments to the Author

Reviewer #1: In this study, the authors analyized the associations between variants in dopaminergic pathway genes with stress resilience. They found that several SNPs and novel mutations in DA pathway were asscociated with stree resilience. I mainly have some concerns about the methods.

1. How many times of tests were corrected in the bonferroni correction? The corrected p value should be reported for each gene site.

2. since the sample size is not big enough, they may want combine the group A and c as group with normal reactions to life stress, and combine group B and D as group wiht with an acute stress reaction to life stress.

3. Did the authors calculate the LD between the SNPs?

4. SNPs detected in the samples should not be regards as point mutations. Are all SNPs in Hardy-Weinberg equilibrium? The HWE is not reported.

5. The results of correlation analysis is interesting, but correlation analysis methods are not clear. Did the done in all samples?

Reviewer #2: It is an interesting study, however, due to the sampe amount is smaller, the author should shrink their conclusion.

Second, the figure 2 should be modified to faciliate the readers to understand.

Third, the author using dopamine pathway to explain the stress reselince inclined to bias, shoud add some discussion about the reciprocal action of dopamine and 5-HT.

Reviewer #3: The authors evaluated the associations between variants in dopaminergic pathway genes with stress resilience in this study. I have a few comments.

1. I noticed that the authors published an article entitled “Low resilience to stress is associated with candidate gene expression alterations in the dopaminergic signalling pathway” in 2018. I think that this article should be cited and I wonder if any relationship between genetic variants and gene expression in dopaminergic signaling pathway under the context of low resilience to stress. The authors should discuss the possibility.

2. For investigation of the association of clinical psychological assessment with genetic variants, only the total score of several types of scales were used for assessment of the relationship, or both of the total score and subscale scores were used? Is any correction method was used in statistical analysis of the association?

3. How about the potential influence of these de novo mutation found in this study? For example, affecting the structure or function of the proteins. The author should provide some clues by searching the relevant database.

4. For the common variants, only allel frequency was analyzed for detection of the potential genetic risk in persons with low resilience to stress, the authors should consider both of the allel frequency and genotype frequency.

6. PLOS authors have the option to publish the peer review history of their article (what does this mean?). If published, this will include your full peer review and any attached files.

Reviewer #1: No

Reviewer #2: No

Reviewer #3: No

---

## [Author Response · Author response to Decision Letter 0]

27 Jun 2021

Dear Editorial Board 

We are writing to answer the reviewers and editors comments about our manuscript entitled" Detection of six novel de novo mutations in individuals with low resilience to psychological stress"

Funding Statement:

The funder provided support in the form of salaries for authors, contributor specialists, and materials. Also, the neuroimaging genetic laboratory of the Arvin gene was used for conducting the main part of the study. Arvin Gene company did not have any additional role in the study design, data collection, and analysis, decision to publish, or preparation of the manuscript. The specific roles of these authors are articulated in the 'author contributions' section."

Conflict of interest:

Arvin Gene's company policies do not alter our adherence to PLOS ONE policies on sharing data and materials. Also, authors and Arvin Gene company as the funder of the project declare that they have no conflict of interests.

Data Availability details: 

Stress problems may consider as a big disadvantage for several careers. Due to the working and insurance situation of participants who admitted from institutional outpatient clinics, especially their information in their institutions' databases, and to avoid potential identification or any other effects on our participants' career we decided to not sharing a de-identified data set, publicly. But to improve the data access for the scientific community we made all the demographic and genetic data available in the Arvin Gene Company Database. These de-identified data are restored and ready to share with any scientist due to his or her written application. Our colleagues are ready to answer the requests by this contact ways: 

Telephone:

+9809355127310 

+982122006664 

Fax 

+982122008049 

 E-mail: 

arvin.gene2020@gmail.com

Here we respond to reviewers comments:

Reviewer #1: In this study, the authors analyzed the associations between variants in dopaminergic pathway genes with stress resilience. They found that several SNPs and novel mutations in the DA pathway were associated with stress resilience. I mainly have some concerns about the methods.

1. How many times tests were corrected in the Bonferroni correction? The corrected p-value should be reported for each gene site.

Thank you, the corrected p values were calculated based on three genotypes of each gene site and the number of detected SNPs in each gene. Revised table with a corrected p-value for each gene site presented in the revised manuscript. 

2. since the sample size is not big enough, they may want to combine the group A and c as a group with normal reactions to life stress and combine group B and D as a group with an acute stress reaction to life stress.

Thank you for your notice; the sample size is a limitation for our study but as it is one of the first studies about genetic bases of stress resilience and the complexity of stress causes we decided to analyze the different types of stress separately. 

3. Did the authors calculate the LD between the SNPs?

Yes, the LD calculations by Plink were mentioned in the revised manuscript. 

4. SNPs detected in the samples should not be regards as point mutations. Are all SNPs in Hardy-Weinberg equilibrium? The HWE is not reported.

Thank you for the notice; yes all detected SNPs are in Hardy-Weinberg equilibrium. HWEs were reported in the revised manuscript. 

5. The results of correlation analysis are interesting, but correlation analysis methods are not clear. Did they do in all samples?

Thank you; yes the correlations were calculated in all samples and separately in each low resilient group (group B and D). Results completed in the revised manuscript. 

Reviewer #2: It is an interesting study, however, due to the sample amount is smaller, and the author should shrink their conclusion.

Thank you we were revised the manuscript discussion and conclusion. 

Second, the figure 2 should be modified to facilitate the readers to understand.

I think you mean table 2, we changed the columns due to presenting the more important information about genetic variations. 

Third, the author using the dopamine pathway to explain the stress resilience inclined to bias should add some discussion about the reciprocal action of dopamine and 5-HT.

Thanks for the notice! Sure we discussed it in the revised manuscript.

Reviewer #3: The authors evaluated the associations between variants in dopaminergic pathway genes with stress resilience in this study. I have a few comments.

1. I noticed that the authors published an article entitled "Low resilience to stress is associated with candidate gene expression alterations in the dopaminergic signaling pathway” in 2018. I think that this article should be cited and I wonder if any relationship between genetic variants and gene expression in dopaminergic signaling pathway under the context of low resilience to stress. The authors should discuss the possibility.

Thank you for your notice; we cited our 2018 publication and discussed the gene expression and genetic variants in the discussion of the revised manuscript. 

2. For investigation of the association of clinical psychological assessment with genetic variants, only the total score of several types of scales were used for assessment of the relationship, or both of the total score and subscale scores were used? Is any correction method was used in the statistical analysis of the association?

Good question, yes both subscales and total scores were assessed; it has been clarified in the revised manuscript. The Bonferroni correction was used for correction. 

3. How about the potential influence of these de novo mutations found in this study? For example, affecting the structure or function of the proteins. The author should provide some clues by searching the relevant database.

Thank you, we discussed the potential functional effects of all de novo mutations in the revised manuscript. 

4. For the common variants, only allele frequency was analyzed for detection of the potential genetic risk in persons with low resilience to stress, the authors should consider both the allele frequency and genotype frequency.

Thank you, yes both of the genotype frequencies and allele frequencies were considered in calculations. The table of genotype frequencies has been reported in the revised manuscript.

---

## [Decision Letter · Decision Letter 1]

4 Aug 2021

Detection of six novel de novo mutations in individuals with low resilience to psychological stress

PONE-D-20-40138R1

Dear Dr. Arvin Haghighatfard,

We’re pleased to inform you that your manuscript has been judged scientifically suitable for publication and will be formally accepted for publication once it meets all outstanding technical requirements.

Kind regards,

Weihua YUE, M.D.

Academic Editor

PLOS ONE

Additional Editor Comments (optional):

Reviewers' comments:

Reviewer's Responses to Questions

**Comments to the Author**

1. If the authors have adequately addressed your comments raised in a previous round of review and you feel that this manuscript is now acceptable for publication, you may indicate that here to bypass the “Comments to the Author” section, enter your conflict of interest statement in the “Confidential to Editor” section, and submit your "Accept" recommendation.

Reviewer #1: All comments have been addressed

2. Is the manuscript technically sound, and do the data support the conclusions?

Reviewer #1: Yes

3. Has the statistical analysis been performed appropriately and rigorously? 

Reviewer #1: Yes

4. Have the authors made all data underlying the findings in their manuscript fully available?

Reviewer #1: Yes

5. Is the manuscript presented in an intelligible fashion and written in standard English?

Reviewer #1: Yes

6. Review Comments to the Author

Reviewer #1: All my comments have been well addressed. This is an interesting study, and I recommend to publish it.

7. PLOS authors have the option to publish the peer review history of their article (what does this mean?). If published, this will include your full peer review and any attached files.

Reviewer #1: **Yes: **Jinsong Tang

---

## [Editor Report · Acceptance letter]

27 Aug 2021

PONE-D-20-40138R1 

Detection of six novel de novo mutations in individuals with low resilience to psychological stress 

Dear Dr. Haghighatfard:

I'm pleased to inform you that your manuscript has been deemed suitable for publication in PLOS ONE. Congratulations! Your manuscript is now with our production department. 

Kind regards, 

on behalf of

Dr. Weihua YUE 

Academic Editor

PLOS ONE